# The Impact of the SMOTE Method on Machine Learning and Ensemble Learning Performance Results in Addressing Class Imbalance in Data Used for Predicting Total Testosterone Deficiency in Type 2 Diabetes Patients

**DOI:** 10.3390/diagnostics14232634

**Published:** 2024-11-22

**Authors:** Mehmet Kivrak, Ugur Avci, Hakki Uzun, Cuneyt Ardic

**Affiliations:** 1Faculty of Medicine, Biostatistics and Medical Informatics, Recep Tayyip Erdogan University, Rize 53100, Türkiye; 2Faculty of Medicine, Endocrinology and Metabolism, Recep Tayyip Erdogan University, Rize 53100, Türkiye; ugur.avci@erdogan.edu.tr; 3Faculty of Medicine, Urology, Recep Tayyip Erdogan University, Rize 53100, Türkiye; hakki.uzun@erdogan.edu.tr; 4Faculty of Medicine, Primary Care Physician, Recep Tayyip Erdogan University, Rize 53100, Türkiye; cuneyt.ardic@erdogan.edu.tr

**Keywords:** SMOTE, imbalance problem, total testosterone, machine learning, ensemble learning

## Abstract

Background and Objective: Diabetes Mellitus is a long-term, multifaceted metabolic condition that necessitates ongoing medical management. Hypogonadism is a syndrome that is a clinical and/or biochemical indicator of testosterone deficiency. Cross-sectional studies have reported that 20–80.4% of all men with Type 2 diabetes have hypogonadism, and Type 2 diabetes is related to low testosterone. This study presents an analysis of the use of ML and EL classifiers in predicting testosterone deficiency. In our study, we compared optimized traditional ML classifiers and three EL classifiers using grid search and stratified k-fold cross-validation. We used the SMOTE method for the class imbalance problem. Methods: This database contains 3397 patients for the assessment of testosterone deficiency. Among these patients, 1886 patients with Type 2 diabetes were included in the study. In the data preprocessing stage, firstly, outlier/excessive observation analyses were performed with LOF and missing value analyses were performed with random forest. The SMOTE is a method for generating synthetic samples of the minority class. Four basic classifiers, namely MLP, RF, ELM and LR, were used as first-level classifiers. Tree ensemble classifiers, namely ADA, XGBoost and SGB, were used as second-level classifiers. Results: After the SMOTE, while the diagnostic accuracy decreased in all base classifiers except ELM, sensitivity values increased in all classifiers. Similarly, while the specificity values decreased in all classifiers, F1 score increased. The RF classifier gave more successful results on the base-training dataset. The most successful ensemble classifier in the training dataset was the ADA classifier in the original data and in the SMOTE data. In terms of the testing data, XGBoost is the most suitable model for your intended use in evaluating model performance. XGBoost, which exhibits a balanced performance especially when the SMOTE is used, can be preferred to correct class imbalance. Conclusions: The SMOTE is used to correct the class imbalance in the original data. However, as seen in this study, when the SMOTE was applied, the diagnostic accuracy decreased in some models but the sensitivity increased significantly. This shows the positive effects of the SMOTE in terms of better predicting the minority class.

## 1. Introduction

### 1.1. Medical Topics

Diabetes Mellitus is a long-term, multifaceted metabolic condition that necessitates ongoing medical management. It is marked by the body’s inability to properly process carbohydrates, fats and proteins, stemming from either a lack of insulin or issues with insulin function [1]. Traditionally, it is mainly categorized into two primary types: Type 1 and Type 2 [2]. Type 2 diabetes constitutes around 90–95% of all diabetes cases. The disease is fundamentally characterized by increasing insulin resistance and gradually decreasing insulin secretion over time, triggered by lifestyle factors in genetically predisposed individuals [3].

Hypogonadism is a syndrome that is a clinical and/or biochemical indicator of testosterone deficiency [4]. Cross-sectional studies have reported that 20–80.4% of all men with Type 2 diabetes have hypogonadism, and Type 2 diabetes can be linked to low testosterone [5]. Some of the clinical features of symptomatic hypogonadism (Figure 1) include erectile dysfunction, loss of libido, depression, irritability, fatigue, anemia, decreased intellectual activity, sleep disturbances, increased abdominal fat, reduced body hair and bone mineral density and lean body mass [6].

Many studies have shown a relationship between testosterone levels and triglycerides (TG) [8], as well as hypertension (HT) [9,10]. Additionally, low testosterone is strongly associated with Type 2 diabetes (T_2_D), with secondary hypogonadism affecting approximately one-third of men with T_2_D [11]. Low testosterone levels, or hypogonadism, have been reported to be linked with insulin resistance, which is the primary pathogenic mechanism underlying T_2_D. Long-term testosterone therapy in hypogonadal men has been shown to prevent the progression of prediabetes and induce remission of T_2_D [12]. These relationships suggest that testosterone plays a role in metabolic syndrome, which encompasses various risk factors, including abdominal obesity, dyslipidemia, hypertension and insulin resistance [13,14]. As a result, testosterone replacement therapy is sometimes considered an additional treatment option for managing metabolic syndrome [15]. Routine measurement of total testosterone levels can present some challenges and limitations. For instance, testosterone levels can fluctuate even within a single day. Therefore, a single measurement may not provide a complete picture of the actual testosterone levels. Due to these reasons, evaluating and interpreting total testosterone levels is complex. Doctors decide to conduct testosterone testing by considering symptoms, the patient’s medical history, and other factors. The diagnosis of TD (testosterone deficiency) requires an evaluation of total testosterone (TT) levels (<300 ng/dL) [16] or free testosterone (FT) levels (<6.5 ng/dL) through blood tests. However, due to high costs, men in the overall population do not routinely monitor their TT and FT levels. This results in a substantial proportion of patients with low testosterone levels who remain undiagnosed and untreated [17].

### 1.2. Artificial Intelligence

Artificial Intelligence (AI) algorithms receive considerable focus in terms of research in the domain of medical diagnosis [18,19]. Clinical decision support systems calculate risk or probability by aggregating multiple predictors, with each predictor being weighted according to its assigned importance. The likelihood of having a disease can be used for urological referral for further tests that focus on the risk of a specific health condition. A literature search related to “testosterone” and “machine learning (ML)” identified many articles [20]. We have evaluated that applying predictive algorithms, such as machine learning and deep learning, to hypogonadism, particularly when the condition results from external factors, presents significant challenges. However, testosterone deficiency (TD) stemming from secondary causes is frequently linked with comorbidities like obesity, metabolic syndrome and systemic diseases, offering a wealth of data that can improve the predictive accuracy of machine learning algorithms. Prediction studies utilizing machine learning (ML) and deep learning (DL) methods have demonstrated high performances. However, two factors make traditional ML approaches adequate for many studies [21]. DL performs poorly with limited data and is therefore better suited for large datasets. In this regard, ML methods are more appropriate; however, identifying the optimal machine learning setup for a specific clinical prediction requires testing a series of procedures. For example, this includes different base or ensemble learner (EL) classifiers, strategies to handle imbalanced learning or the use of various measures for evaluating classification performance. Many studies have used and compared the ML technique [22]. Some challenges are hard to address with a single machine-learning classifier, and the best approach is to use an ensemble-based classifier that integrates multiple models to enhance prediction performance [23]. Ensemble-based classifiers have proven effective in many clinical branches, such as Alzheimer’s diagnosis, breast cancer, and cardiovascular diseases [24]. When applied to small datasets, there are advantages in terms of increased performance and multiple comparison options among the methods due to the tendency to explore various hypotheses in training data prediction and the broad combination of models [25]. In ML methods, data preprocessing is an important step. Depending on the ratio of negative to positive samples, imbalanced data may need to be preprocessed, as traditional algorithms tend to consider minority observations as noise. In this context, imbalanced data can lead to biased results in predictive modeling. Addressing the class imbalance problem at the data level is a crucial step in the preprocessing phase [26].

This study presents an analysis of the use of ML and EL classifiers in predicting testosterone deficiency. In our study, we compared optimized traditional ML classifiers and three EL classifiers using grid search and stratified k-fold cross-validation. We used the SMOTE method for the class imbalance problem. Finally, we compared multiple performance metrics of the base and EL classifiers.

## 2. Methods

The working steps are given in Figure 2 below. These steps involve dataset acquisition and splitting, solution of the class balance problem (SMOTE), base classifiers, ensemble classifiers (second-level classifiers), stratified k-fold cross-validation, grid searching, and accuracy analyses.

### 2.1. Dataset

This database contains 3397 patients for the assessment of testosterone deficiencies. Among these patients, 1886 patients with Type 2 diabetes were included in the study. The study was granted approval by the Research Ethics Committee of the State University of Feira de Santana in Bahia, Brazil, with the ethical approval code 3.057.301 [27]. The variables used in the study, their roles and definitions are given in Table 1. Participants aged between 45 and 85 years, with a mean age of 62.5, were included in the study. The participants’ mean TG was 170.0; the mean HDL was 45.4 and the mean AC was 102.0. While 61.9% of the participants had HT, 38.1% did not have the disease.

### 2.2. Data Preprocessing

In the data preprocessing stage, firstly, outlier/excessive observation analyses were performed with local outlier factor (LOF) (Figure 3), and missing value analyses were performed with random forest. LOF is an unsupervised outlier detection method. This algorithm assesses the uniqueness of each event while focusing on the distance to its k-nearest neighbors. Because the LOF algorithm does not make any assumptions about data distributions, it can detect outliers independently of the data distribution. The core concept is that the density surrounding an outlier object markedly differs from the density around its neighboring points [28]. Then, the class balance problem was addressed. We divided the testosterone level into two classes: (a) 0 (T < 300 ng/dL) and (b) 1 (T ≥ 300 ng/dL). Regarding data splitting, we allocated 30% of the data solely for the testing phase and then applied stratified k-fold cross-validation (k = 10) on the remaining 70% of the data. The test set provided independent validation, demonstrating the model’s proficiency in handling data it has not encountered before. The operations in this process were calculated with the loffactor function in the R program dprep package. 

#### The Problem of Class Imbalance

Figure 4 is an illustration of class imbalance; green shows patients with testosterone deficiency (TD) (T < 300 ng/dL) and blue shows patients with normal testosterone levels (T ≥ 300 ng/dL). Datasets are imbalanced when the distribution of classes is unequal [29].

ML algorithms often yield poor classification results on imbalanced data. There are several ways to address class imbalance [30]. The Synthetic Minority Over-sampling Technique (SMOTE) is a method for generating synthetic samples of the minority class. It typically outperforms simple oversampling and is commonly employed in various applications [31]. The SMOTE method generates a synthetic sample by linearly combining two samples from the minority class (*X_i_* and *X_j_*) as follows:
(1)*X_new_* = *X_i_* + (*X_j_*− *X_i_*) * α

For the new artificial instance *X_new_* of the minority class, a sample *X_i_* is selected randomly. Then, *X_i_* is chosen randomly among the five nearest neighbors of *X_i_* from the minority class based on the Euclidean distance [32]. The parameter α takes a random float value in the range (0, 1) [33]. This research used the SMOTE function in the R program (4.4.2) open source Smotefamily package.

### 2.3. Statistical Analysis

Before statistical analysis, a normality test (Kolmogorov–Smirnov) was applied to the dataset, and not all variables met the normality assumption. Descriptive statistics were given as Mean ± Standard Deviation and Median/(IQR) for continuous data, while they were given as count and percentage for categorical variables. For the significance test of the difference in group categories (testosterone deficiency), the Mann–Whitney U Test was used for continuous data and the Chi-Square Test for independent groups was used for categorical data. All statistical analyses were performed with the IBM-SPSS 25.0 program.

### 2.4. Base Classifiers

In machine learning methods, the use of base classifiers with low correlation with each other enables comprehensive comparisons [34]. Four basic classifiers, namely multilayer perceptron, random forest, Extreme Learning Machine and logistic regression, were used as first-level classifiers.

#### 2.4.1. Multilayer Perceptron (MLP)

MLP is one of the most widely utilized artificial neural network models. It has been extensively studied, leading to the development of numerous learning algorithms. MLP is a type of forward-feeding, fully connected neural network that transforms an input dataset into a corresponding output set by fine-tuning the weights between its internal nodes [35]. The input layer contains n input variables *X* = (*x*_1_, *x*_2_,…, *x_n_*) and the output layer contains *Y* = (*y*_1_, *y*_2_, …, *y_m_*). The overall count of parameters in an MLP can be determined by [36] as follows:(2)$$n*h_{1}+\sum_{k=1}^{Nh-1}h_{k}*h_{k+1}+h_{N_{h}*n}$$
where the number of hidden nodes *h_i_* in the *i*th layer is *N_h_*. Longer computational times are required to optimize an MLP when *N_h_* and *h_k_* are higher [37].

#### 2.4.2. Random Forest (RF)

RF is an ensemble method that combines multiple decision tree classifiers. It can be seen as an enhanced form of the bagging technique. The RF algorithm works as follows: each decision tree in the forest is created using the bootstrap re-sampling method. This technique allows for the generation of multiple datasets by generating new samples with replacements from the original dataset, regardless of its size. This approach, known as the “Bootstrap Re-sampling Method”, enables the extraction of more information from the data. Different samples are then generated by selecting subsets of the data. The RF model aggregates the class predictions from all the decision trees to determine the most accurate class prediction. In RF, a subset of m variables is randomly chosen from the entire set of variables for each tree, and this subset remains constant for each tree. Typically, the number of variables m is chosen as $\sqrt{p}$ (where *p* is the total number of variables) [38].

#### 2.4.3. Extreme Learning Machine (ELM)

Extreme Learning Machine (ELM) is a batch regression algorithm designed to train the weights of a Single-Hidden Layer Feedforward Network (SLFN). An SLFN is a type of artificial neural network (ANN) featuring three layers: an input layer, a hidden layer and an output layer.
(3)$$f\left(x\right)=\sum_{e=1}^{E}w_{e}\phi (\sum_{d=1}^{D}v_{ed}x_{d}),$$
(4)$$f(x)=\sum_{e=1}^{E}w_{e}\phi \left(z_{e}\right),\;$$
where *z_e_* = $\sum_{d=1}^{D}v_{ed}x_{d\;}$ is the input to the activation function ϕ, which is often chosen to be sigmoid or hyperbolic [39].

#### 2.4.4. Logistic Regression (LR)

Linear models consist of one or more independent variables that establish a relationship with a dependent response variable. In the context of ML, when qualitative or quantitative input features are mapped to a target variable that we aim to predict, such as in financial, biological, or sociological data, this approach is called supervised learning, provided that the labels are known. LR is among the best frequently utilized linear statistical models for discriminant analysis.
(5)$$y_{i}=\beta_{0}+\beta_{1}X_{1}+\beta_{2}X_{2}+\ldots +\beta_{n}X_{n}$$

*y_i_* = dependent variable, *β*_0_ = constant, *β_n_* = *n*. beta coefficient and *X_n_* = *n*. independent variables. In logistic regression, the response variable is quantitative. Specifically, the response variable represents the logarithm of the odds of being classified into the *i*th group in a binary or multi-class response situation [40].

### 2.5. Ensemble Classifiers

A boosting algorithm is a method for creating strong classifiers from weak ones with minimal training error. It works by combining a group of weak classifiers using a simple majority vote approach [41].

#### 2.5.1. Adaboost

AdaBoost is among the best widely used algorithms for building a strong classifier by linearly combining individual classifiers. During the training process, the individual classifiers are chosen to minimize errors at each iteration. AdaBoost offers a straightforward and effective way to create ensemble classifiers [42].

#### 2.5.2. XGBoost

XGBoost is fundamentally a decision tree boosting algorithm. Boosting is an ensemble learning technique that involves building multiple models in sequence, with each new model being designed to address the shortcomings of the previous one. In tree boosting, every new model added to the ensemble is a decision tree. We will explain how to build a decision tree model and how this process can be extended to generalized gradient boosting using the XGBoost algorithm [43].

#### 2.5.3. Stochastic Gradient Boosting (SGB)

SGB is an ensemble learning algorithm that combines boosting with decision trees. It makes predictions by assigning weights to the ensemble members of all trees in the model [44].

### 2.6. Parameter Optimization

#### 2.6.1. k-Fold Cross-Validation

k-fold cross-validation splits the dataset into k equally sized folds while preserving the original ratio of positive and negative instances. In each iteration, k-1 folds are used for training while the remaining folds are used for testing. The final result is the average accuracy metric across all testing bins. This method provides more realistic results compared to the standard train/test split, particularly for parameter optimization, as it utilizes multiple aspects of the data and reduces variance [34].

#### 2.6.2. Grid Search

Grid search is a systematic search method for the hyperparameter space, generating all possible combinations regardless of the effects of elements in the optimization process. All parameters have an equal chance of influencing this process (Table 2). While this method provides certain guarantees, it also has significant disadvantages. For instance, in an optimization with many parameters, each having several values, it creates a large variety of combinations, leading to extensive computational effort and time consumption [36]. The R program crosval package was used for k-fold cross-validation and grid searching.

### 2.7. Performance Metrics

Diagnostic accuracy [45], sensitivity [46], specificity [46], F1 score [47], positive predict value, and negative predict value [48] were used in the performance metrics. Detailed information is given in Table 3.

## 3. Results

According to Table 4, the testosterone level group categories showed no statistically significant differences ((normal and TD) according to the age variable (*p* = 0.455)). There were significant differences in the TG, HT, HDL and AC variables according to the group categories (*p* < 0.001).

After the SMOTE method was applied to solve the class balance problem in testosterone levels, the normal (T ≥ 300 ng/dL) and TD (T < 300 ng/dL) category distributions reached a balanced structure (Figure 5).

The results in Table 5 show the performance of the classifiers in a study to predict total testosterone deficiency in patients with Type 2 diabetes. The table presents the evaluations performed with original and SMOTE data using four basic classifiers. After using the SMOTE method, while the diagnostic accuracy decreased in all base classifiers except ELM, sensitivity values increased in all classifiers. Similarly, while the specificity values decreased in all classifiers, the F1 score increased. In the MLP and ELM classifiers, a positive predictive value could not be calculated in the original data, but calculations were made after the SMOTE. In other classifiers, positive predictive value decreased after the SMOTE. The negative predictive value increased after the SMOTE in all base classifiers. The classification diagram for the original and SMOTE data using the base classifiers (training data) is presented in Figure 6. The RF classifier gave more successful results with the training dataset.

Table 6 presents the evaluations performed using three ensemble classifiers on the original and SMOTE data. After the SMOTE, the diagnostic accuracy decreased in all ensemble classifiers and the sensitivity values increased in all classifiers. The specificity values decreased in two classifiers except SGB, but the F1 score increased in the XGBoost classifier. The negative predictive values increased after the SMOTE in all ensemble classifiers. The most successful ensemble classifier in the training dataset was the ADA classifier in the original data and the post-SMOTE data. The classification diagram for the original and SMOTE data using the ensemble classifiers (training data) is presented in Figure 7.

According to Table 7, which was created with testing data, MLP shows high specificity with the original data but also an unstable performance, as the sensitivity is very low. With the SMOTE, the sensitivity increases while the specificity decreases; thus, the F1 score becomes calculable. RF provides good stability with the original data. With the SMOTE, sensitivity increases, but overall accuracy drops slightly. LR exhibits high specificity but low sensitivity with the original data. With the SMOTE, sensitivity increases, but accuracy and specificity decrease. ELM somewhat reduces the instability by increasing sensitivity with the SMOTE, but accuracy and specificity are reduced. ADA shows stable performance with the original data and improved sensitivity with the SMOTE. XGBoost maintains its overall accuracy while improving sensitivity with the SMOTE. While SGB shows a balanced performance with the original data, the SMOTE provides increased sensitivity but decreased accuracy. As a result of this analysis, we can say that XGBoost is the most suitable model for the intended use of evaluating model performance. XGBoost, which exhibits a balanced performance, especially when the SMOTE is used, may be preferable in regard to correcting class imbalance.

## 4. Discussion

In recent years, the use of machine learning algorithms in the field of medicine has significantly increased [49]. In this study, we aimed to predict total testosterone deficiency in patients with Type 2 diabetes by using machine learning and ensemble learning methods on original data with class imbalance and data after using the SMOTE. LR is a commonly utilized method in medicine and is frequently cited in the literature. For instance, the study by Hastie et al. (2009) reported that, while LR achieves high accuracy and specificity, it often shows lower sensitivity, particularly when addressing imbalanced datasets [50]. Additionally, in this study, LR achieved 77% diagnostic accuracy and 98% specificity with the original dataset, while its sensitivity remained only 19%. This indicates that the LR model is limited in detecting the positive class, and this observation aligns with findings in the literature concerning imbalanced datasets. The RF model is generally known as a robust classifier and performs well on imbalanced datasets. In a study by Liu et al. (2009), it was noted that RF is particularly effective when used in conjunction with techniques like the SMOTE to address class imbalance [51]. In this study, the RF model achieved 76% diagnostic accuracy and 44% sensitivity with the original dataset, and, after applying the SMOTE, its sensitivity increased to 58%. This finding, where the SMOTE enhances RF performance, is consistent with the results reported by Liu and colleagues. ELM and MLP are generally reported in the literature to perform well with complex datasets, but their sensitivity may be low with imbalanced datasets. In the study by Huang et al. (2012), it was also noted that ELM is a strong model, particularly for multi-class classification problems, but its performance can decline in situations of data imbalance compared to other methods. In this context, the results presented in the table align with the literature regarding ELM’s sensitivity to imbalanced datasets [52]. Similarly, in this study, ELM showed 34% sensitivity with the original dataset, which increased to 49% with the SMOTE. However, the overall diagnostic accuracy dropped from 76% to 69% across both datasets, indicating that addressing imbalance with the SMOTE does not necessarily improve all metrics. Ensemble learning algorithms such as ADA, XGBoost and SGB are generally known for their strong performance with imbalanced datasets. Freund and Schapire (1997) demonstrated that AdaBoost improves sensitivity, especially with imbalanced datasets, by iteratively reducing classification errors [53]. Additionally, in this study, AdaBoost increased its sensitivity from 42% to 61% when used with the SMOTE. However, a decrease in diagnostic accuracy and specificity was observed; this indicates, as noted in the literature, that addressing data imbalance does not always have a positive impact on all metrics. In recent years, XGBoost has emerged in the literature as a model with strong performance in regard to large datasets and imbalanced classes. In their study, Chen and Guestrin (2016) reported that XGBoost is particularly effective in classification problems and achieves successful results when used with the SMOTE in situations of data imbalance [54]. In this study, XGBoost achieved a 75% diagnostic accuracy with the original dataset; with the SMOTE, this accuracy changed to 73%, while sensitivity increased to 52%. This result is consistent with the findings of Chen and Guestrin, demonstrating that XGBoost can provide a balanced performance with imbalanced datasets.

In the clinical context, the trade-off between sensitivity and specificity is important, especially in finding the right balance in diagnostic tests. Increasing sensitivity can increase false positives and decrease specificity; similarly, increasing specificity can lead to missing some positive cases. In the clinical context, having high sensitivity is important as it helps to avoid missing individuals with the disease. On the other hand, when specificity is higher, the false positive rate is reduced, ensuring that only individuals who really need it are treated. For example, a study by Sun et al. (2021) discusses cost-sensitive learning and sample augmentation techniques such as the SMOTE in terms of imbalanced datasets. It also investigates how to optimize the balance between sensitivity and specificity while addressing class imbalance [55]. In a study by Vickers et al. (2006), in addition to analyzing the trade-off between sensitivity and specificity in evaluating medical prediction models, balancing this trade-off with decision curve analysis methods was also discussed [56]. 

## 5. Conclusions

The SMOTE is used to correct the class imbalance in the original data. However, as seen in this study, when the SMOTE was applied, the diagnostic accuracy decreased in some models but the sensitivity increased significantly. This shows the positive effects of the SMOTE in terms of better predicting the minority class. RF and ELM models showed a higher increase in sensitivity and overall performance after SMOTE application, indicating that these models can be more effective in imbalanced datasets. If total testosterone deficiency is chosen as the reference group (positive class), the choice of meaningful metrics depends on the purpose of the prediction and its clinical significance. Sensitivity, F1 score, and positive predictive value (PPV) will be the most meaningful metrics when you choose total testosterone deficiency as the reference group because these metrics focus on the importance of correctly identifying individuals with deficiency, which is one of the most critical points for clinical decisions. Our study was conducted only on patients with Type 2 diabetes; therefore, the generalizability of the results obtained to different populations is limited. The demographic characteristics of the patients in our dataset are limited, which affected the performance of the model. For example, in a study by Doorn et al. (2021), it is discussed that health outcomes vary in patients with different types of diabetes and that the performance of the models may vary depending on the population. This limitation, while limiting the generalizability of the results, reveals the need for a broader analysis on the potential application areas of the model. The effect of the features in the dataset (age, gender, other health conditions, etc.) on the classifier performance may cause the model to be trained only for certain features, which may lead to the model being less generalizable to certain patient groups. Therefore, additional studies with larger datasets obtained from different populations are needed in the future [55]. Although synthetic data generated with the SMOTE method increased the sensitivity and positive classification performance of the model, it is important to note that this method may not fully reflect the natural variation in the dataset in some cases. The use of synthetic data may cause the model to be optimized only for the current population and carries the risk of not providing consistent results with other datasets. In a study by Goncalves et al. (2020), the benefits and potential limitations of synthetic data are examined, particularly the fact that the generalization capacity of models trained using synthetic data may be limited compared to models trained with natural data [57].

In conclusion, this study shows that classifier performance is highly sensitive to data imbalance and that techniques like the SMOTE play a crucial role in addressing this imbalance. Specifically, the XGBoost model exhibited the highest performance in sensitivity and diagnostic accuracy when combined with the SMOTE. These results are in line with findings from similar studies in the literature. XGBoost, which provides balanced performance, especially when used in combination with the SMOTE, can be preferred for correcting class imbalance. Future research could explore the application of such models in the early diagnosis of various diseases (e.g., cardiovascular diseases, endocrine disorders). Additionally, the integration of these models with real-time clinical data streams could support healthcare providers in making faster and more reliable decisions. Another important area for future research is the use of larger and more diverse datasets. The dataset used in this study represents a specific population, which may limit the generalizability of the model. Research involving data from different demographic groups (e.g., age, gender, ethnicity) and geographic regions could improve the model’s generalizability and test its validity in a broader patient population. Utilizing larger datasets could also help examine the effects of different health conditions on the prediction of testosterone deficiency, contributing to the development of more comprehensive strategies for clinical applications. Moreover, other techniques to address class imbalance beyond the SMOTE could be investigated. For instance, various data augmentation strategies, next-generation deep learning techniques, or reinforcement learning methods might offer better performance and could be tested on different healthcare data. Finally, this study also highlights the technical and ethical challenges that must be considered when applying machine learning algorithms in clinical settings. In clinical environments, it is crucial that these algorithms produce accurate and reliable results while still being user-friendly and safe for healthcare professionals to use. 

## Figures and Tables

**Figure 1 diagnostics-14-02634-f001:**
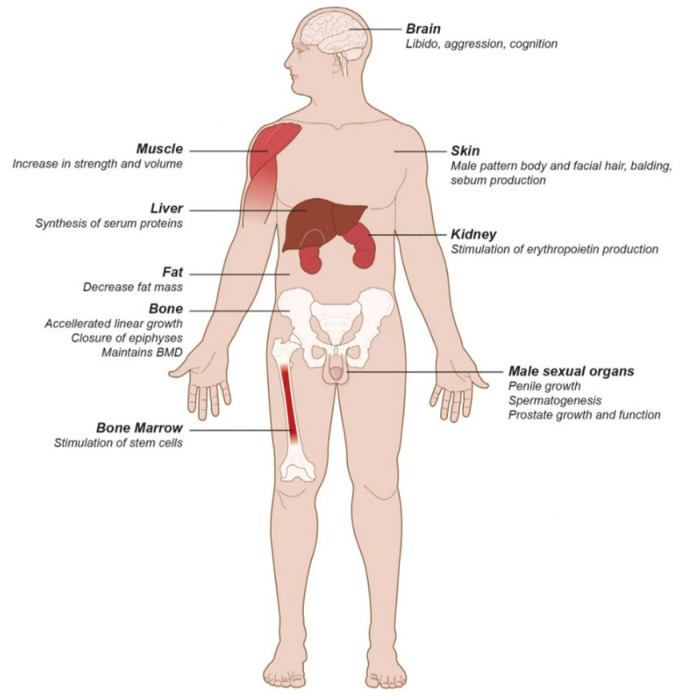
Testesterone target organs [7].

**Figure 2 diagnostics-14-02634-f002:**
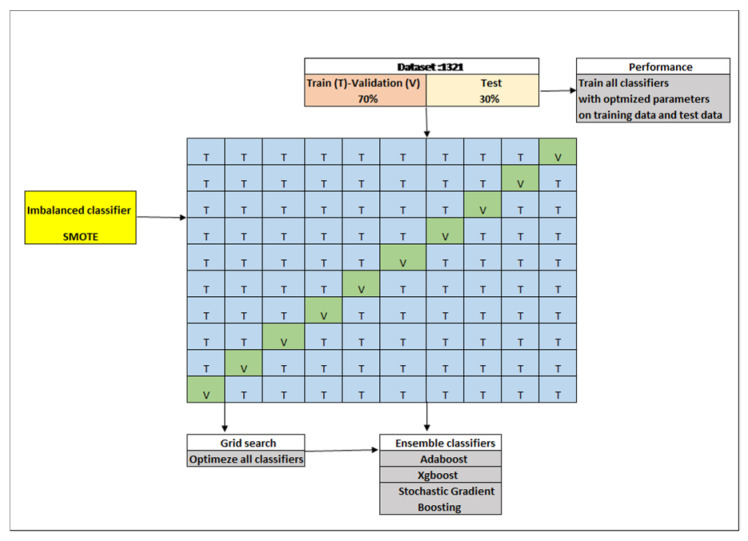
The working step.

**Figure 3 diagnostics-14-02634-f003:**
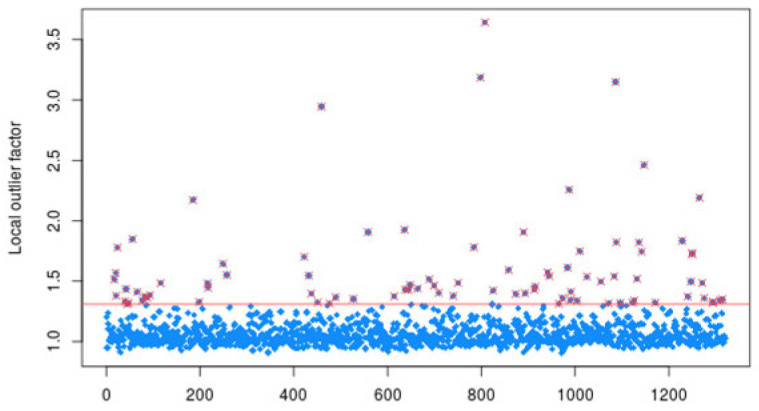
Outlier/excessive observation analyses with local outlier factor. The observations shown in blue in the figure are values within normal limits. The values above the red line are outliers.

**Figure 4 diagnostics-14-02634-f004:**
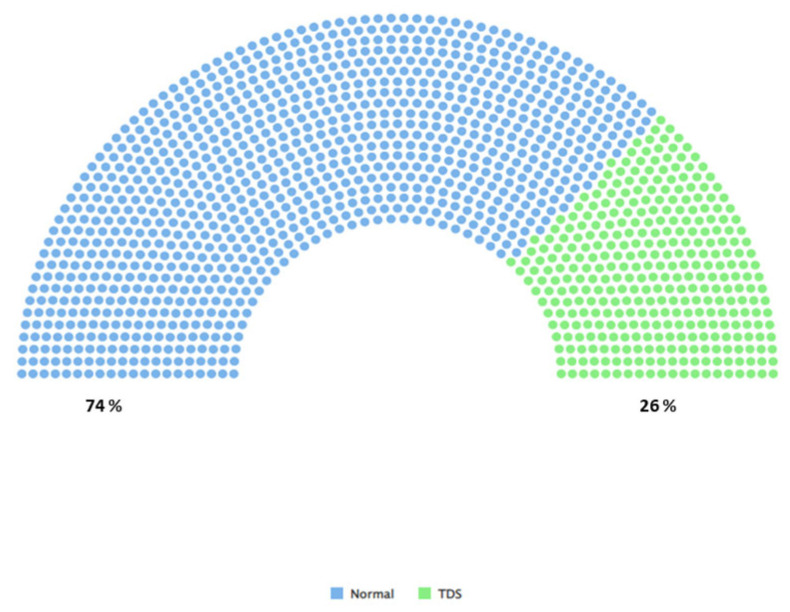
Illustration of class imbalance.

**Figure 5 diagnostics-14-02634-f005:**
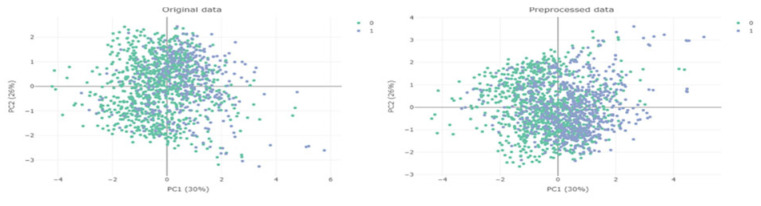
Original and preprocessed (SMOTE) data. Blue color normal individuals and green color testosterone deficiency (TD) individuals.

**Figure 6 diagnostics-14-02634-f006:**
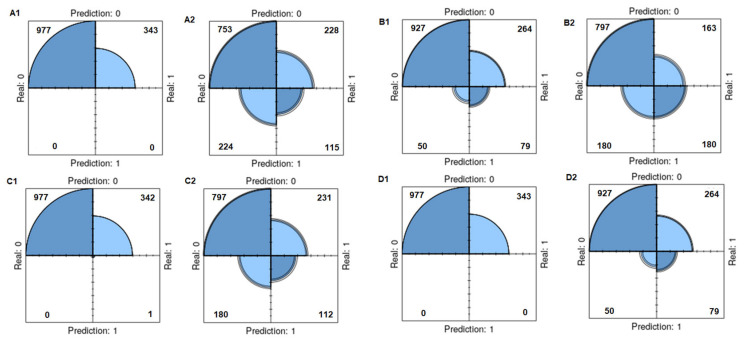
Classification diagram for original and SMOTE data using base classifiers (training data). (**A1**): Original data of MLP, (**A2**): SMOTE data of MLP, (**B1**): Original data of RF, (**B2**): SMOTE data of RF, (**C1**): Original data of LR, (**C2**): SMOTE data of LR, (**D1**): Original data of ELM, (**D2**): SMOTE data of ELM.

**Figure 7 diagnostics-14-02634-f007:**
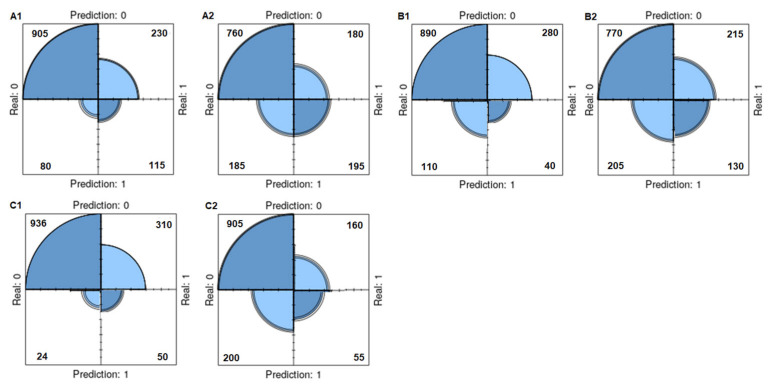
Classification diagram for original and SMOTE data using base classifiers (training data). (**A1**): Original data of ADA, (**A2**): SMOTE data of ADA, (**B1**): Original data of XGBoost, (**B2**): SMOTE data of XGBoost, (**C1**): Original data of SGB, (**C2**): SMOTE data of SGB.

**Table 1 diagnostics-14-02634-t001:** The Variables Used In The Study.

Variable	Role	Description	Range	Descriptive Statistics
Age	Input	Age in years	45–85	$\bar{X}$ = 62.5; S_d_ = 9.7
TG	Input	Triglycerides (mg/dL)	12–980	$\bar{X}$ = 170.0; S_d_ = 99.7
HT	Input	Hypertension	Yes/No	Yes = 61.9%; No = 38.1%
HDL	Input	High-density lipoprotein (mg/dL)	24–102	$\bar{X}$ = 45.4; S_d_ = 11.0
AC	Input	Abdominal Circumference (cm)	43–198	$\bar{X}$ = 102.0; S_d_ = 10.9
T	Output	Testosterone (ng/dL)	47–1375	$\bar{X}$ = 552.9; S_d_ = 182.1

$\bar{X}$ = Arithmetic Mean; S_d_ = Standard Deviation.

**Table 2 diagnostics-14-02634-t002:** Grid Search Values for Each Classifier.

Model	Parameter	Values
RF	Bootstrap	(True, False)
	oob_score	(True, False)
	max_depth	3, 4, 5, 6, 7
	n_estimators	50, 100, 150, 200, 250
	min_samples_split	2, 3, 4, 5
	max_leaf_nodes	None, 2, 3, 4
MLP	hidden_layer_sizes	(10, 10), (15, 15), (20, 10), (20, 15)
	Activation	tanh, relu
	learning_rate	0.01, 0.001
	max_iter	200, 400, 600
	Solver	Lbfgs, sgd, adam
ELM	hidden_layer_sizes	(10,10)
	Activation	sigmoid
	learning_rate	0.01, 0.001
	max_iter	200, 400, 600
	Solver	adam
LR	C	0.5, 1, 2, 3, 4, 5, 6
	Penalty	L1, L2, elasticnet
	Solver	Newton-cg, lbfgs, saga
	max_iter	50, 100, 200
	class_weight	Balanced, None
ADA	DT max_depth	None, 2, 3, 4
	DT min_samples_split	2, 3, 4
	DT max_leaf_nodes	None, 2, 4
	DT max_features	None, 3, 5
	n_estimators	300, 400, 500, 600
	learning_rate	0.1, 0.01
XGBoost	min_child_weight	1, 3, 5, 7, 10
	Gamma	1, 3, 5, 7, 10
	colsample_bytree	0.4, 0.5, 0.6
	reg_alpha	0, 0.2, 0.3
	max_depth	4, 5, 6
	Subsample	0.6, 0.7, 0.8
	n_estimators	100, 200, 300, 400, 500
	learning_rate	0.1, 0.01
SGB	min_child_weight	1, 3, 5, 7, 10
	max_depth	1,3,5
	max_features	0.2,0.4
	n_estimators	300, 600
	learning_rate	0.1, 0.01

**Table 3 diagnostics-14-02634-t003:** Performance Metrics.

Performance Metric	Formula	Referance
D.Accuracy	(TP + TN)/(TP + FP + FN + TN)	[45]
Sensitivity	TP/(TP + FN)	[46]
Specificity	TN/(FP + TN)	[46]
F1 Score	TP/TP + 0.5(FP + FN)	[47]
P.Predict Value	TP/(TP + FP)	[48]
N.Predict Value	TN/(TN + FN)	[48]

TP = true positive; TN = true negative; FP = false positive; FN = false negative.

**Table 4 diagnostics-14-02634-t004:** Statistical Analysis for Variables.

Group	Normal	TD		
Variable	Descriptive Statistics		Test Statistics	*p*-Value
Age	$\bar{X}$ = 62.4; S_d_ = 9.8; Med = 62.0; IQR = 15.0	$\bar{X}$ = 62.8; S_d_ = 9.5; Med = 61.0; IQR = 14.0	163,019	0.455 *
TG	$\bar{X}$ = 157.0; S_d_ = 86.3; Med = 139.0; IQR = 91.0	$\bar{X}$ = 207.0; S_d_ = 123.0 Med = 171.0; IQR = 119.0	116,978	<0.001 *
HT	Count = 977; Percent = 74	Count = 343; Percent = 26	15.7	<0.001 **
HDL	$\bar{X}$ = 46.3; S_d_ = 10.8; Med = 44.0; IQR = 13.0	$\bar{X}$ = 43.1; S_d_ = 11.1; Med = 41.0; IQR = 12.0	134,120	<0.001 *
AC	$\bar{X}$ = 99.9; S_d_ = 10.1; Med = 98.0; IQR = 11.0	$\bar{X}$ = 107.0; S_d_ = 13.0; Med = 62.0; IQR = 15.0	92,879	<0.001 *

$\bar{X\;}$ = Arithmetic Mean; S_d_ = Standard Deviation; Med = Median; IQR = Interquartile Range; * = Mann–Whitney U Test; ** = Chi-Square Test for independent groups; TD = Testosterone Deficiency.

**Table 5 diagnostics-14-02634-t005:** Performance Metrics Original and SMOTE Data for Base Classifiers (Training Data).

Classifiers	Metrics					
Base Classifiers	Diagnostic Accuracy	Sensitivity	Specificity	F1 Score	Positive Predictive Value	Negative Predictive Value
MLP						
Original Data	0.74	NA	1.00	NA	NA	0.74
SMOTE	0.66	0.34	0.77	0.34	0.34	0.77
RF						
Original Data	0.76	0.23	0.95	0.34	0.61	0.78
SMOTE	0.74	0.53	0.82	0.51	0.50	0.83
LR						
Original Data	0.74	0.03	1.00	0.06	1.00	0.74
SMOTE	0.69	0.33	0.82	0.35	0.38	0.78
ELM						
Original Data	0.74	NA	1.00	NA	NA	0.74
SMOTE	0.76	0.23	0.95	0.34	0.61	0.78

NA: Not applicable.

**Table 6 diagnostics-14-02634-t006:** Performance Metrics Original and SMOTE Data for Ensemble Classifiers (Training Data).

Classifiers	Metrics					
Ensemble Classifiers	Diagnostic Accuracy	Sensitivity	Specificity	F1 Score	Positive Predictive Value	Negative Predictive Value
ADA						
Original Data	0.77	0.33	0.92	0.43	0.60	0.80
SMOTE	0.73	0.52	0.81	0.52	0.52	0.81
XGBoost						
Original Data	0.71	0.13	0.89	0.17	0.27	0.76
SMOTE	0.68	0.38	0.79	0.38	0.39	0.79
SGB						
Original Data	0.75	0.14	0.68	0.23	0.68	0.75
SMOTE	0.73	0.26	0.82	0.24	0.22	0.85

**Table 7 diagnostics-14-02634-t007:** Performance Metrics Original and SMOTE Data for Base-Ensemble Classifiers (Testing Data).

Classifiers	Metrics					
Base Classifiers	Diagnostic Accuracy	Sensitivity	Specificity	F1 Score	Positive Predictive Value	Negative Predictive Value
MLP						
Original Data	0.74	0.01	1.00	NA	NA	0.74
SMOTE	0.70	0.42	0.84	0.48	0.56	0.75
RF						
Original Data	0.76	0.44	0.87	0.49	0.54	0.82
SMOTE	0.68	0.58	0.85	0.54	0.51	0.85
LR						
Original Data	0.77	0.19	0.98	0.30	0.73	0.78
SMOTE	0.71	0.31	0.89	0.42	0.63	0.86
ELM						
Original Data	0.76	0.34	0.91	0.42	0.57	0.80
SMOTE	0.69	0.49	0.82	0.47	0.38	0.89
Ensemble Classifiers	Diagnostic Accuracy	Sensitivity	Specificity	F1 Score	Positive Predictive Value	Negative Predictive Value
ADA						
Original Data	0.72	0.42	0.85	0.48	0.56	0.76
SMOTE	0.67	0.61	0.79	0.52	0.51	0.82
XGBoost						
Original Data	0.75	0.41	0.87	0.47	0.52	0.81
SMOTE	0.73	0.52	0.81	0.51	48.00	0.88
SGB						
Original Data	0.73	0.34	0.87	0.40	0.47	0.79
SMOTE	0.69	0.48	0.77	0.44	0.42	0.83

## Data Availability

Publicly available archived datasets analyzed or generated during the study are available in [github] at [https://github.com/osmarluiz/Testosterone-Deficiency-Dataset] (accessed on 17 September 2024).

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
