# Peer review of "The Impact of the SMOTE Method on Machine Learning and Ensemble Learning Performance Results in Addressing Class Imbalance in Data Used for Predicting Total Testosterone Deficiency in Type 2 Diabetes Patients"

_diagnostics, 2024, doi:10.3390/diagnostics14232634_

Round 1
Reviewer 1 Report
Comments and Suggestions for Authors
The article presents a comprehensive study on the application of machine learning (ML) and ensemble learning (EL) techniques to predict total testosterone deficiency in patients with type 2 diabetes, addressing the critical issue of class imbalance using the Synthetic Minority Oversampling Technique (SMOTE). The authors, provide a detailed methodology, including data preprocessing, classifier selection, and performance evaluation metrics. Congratulate the authors for the work done, which contains valuable information. The Topic is novel and the article is clear and presented in a well-structured manner. I have the following suggestions:
1. The article could benefit from a more thorough exploration of the study's limitations. Specifically, aspects such as potential biases within the dataset, the applicability of the findings to different populations, and the consequences of utilizing synthetic data should be considered.
2. The authors mention that diagnostic accuracy decreased in some models after applying SMOTE, yet sensitivity increased significantly. A more in-depth analysis of the trade-offs between sensitivity and specificity, particularly in a clinical context, would strengthen the discussion.
3. The article could include a section on future research directions, suggesting how the findings could inform clinical practice or lead to further studies exploring other machine learning techniques or larger, more diverse datasets.

Author Response
Comment 1: The article could benefit from a more thorough exploration of the study's limitations. Specifically, aspects such as potential biases within the dataset, the applicability of the findings to different populations, and the consequences of utilizing synthetic data should be considered.
Reply: Our study was conducted only on patients with type 2 diabetes; therefore, the generalizability of the results obtained to different populations is limited. The demographic characteristics of the patients in our dataset are limited, which affected the performance of the model. For example, in a study by Doorn et al. (2021), it is discussed that health out-comes vary in patients with different types of diabetes and that the performance of the models may vary depending on the population. This limitation, while limiting the generalizability of the results, reveals the need for a broader analysis on the potential application areas of the model. The effect of the features in the dataset (age, gender, other health conditions, etc.) on the classifier performance may cause the model to be trained only for certain features, which may lead to the model being less generalizable to certain patient groups. Therefore, additional studies with larger datasets obtained from different populations are needed in the future (55). Although synthetic data generated with the SMOTE method in-creased the sensitivity and positive classification performance of the model, it is important to note that this method may not fully reflect the natural variation in the dataset in some cases. The use of synthetic data may cause the model to be optimized only for the current population and carries the risk of not providing consistent results on other datasets. In a study by Goncalves et al. (2020), the benefits and potential limitations of synthetic data are examined, particularly the fact that the generalization capacity of models trained using synthetic data may be limited compared to models trained with natural data (56). The relevant text was added to the conclusion section.
comment 2: The authors mention that diagnostic accuracy decreased in some models after applying SMOTE, yet sensitivity increased significantly. A more in-depth analysis of the trade- offs between sensitivity and specificity, particularly in a clinical context, would strengthen the discussion.
Reply: In the clinical context, the trade-off between sensitivity and specificity is important, especially in finding the right balance in diagnostic tests. Increasing sensitivity can in-crease false positives and decrease specificity; similarly, increasing specificity can lead to missing some positive cases. In the clinical context, having high sensitivity is important to avoid missing individuals with the disease. On the other hand, when specificity is higher, the false positive rate is reduced, ensuring that only individuals who really need it are treated. For example, in a study by Sun et al. (2021), discusses cost-sensitive learning and sample augmentation techniques such as SMOTE for imbalanced datasets. It also investigates how to optimize the balance between sensitivity and specificity while addressing class imbalance (55). In a study by Vickers et al. (2006), in addition to analyzing the trade-off between sensitivity and specificity in evaluating medical prediction models, we also indicated how to balance it with decision curve analysis methods (56). The relevant text was added to the discussion section.
Comment 3: The article could include a section on future research directions, suggesting how the findings could inform clinical practice or lead to further studies exploring other machine learning techniques or larger, more diverse datasets.
Reply: Future research could explore the application of such models in the early diagnosis of various diseases (e.g., cardiovascular diseases, endocrine disorders). Additionally, the in-tegration of these models with real-time clinical data streams could support healthcare providers in making faster and more reliable decisions. Another important area for future research is the use of larger and more diverse datasets. The dataset used in this study rep-resents a specific population, which may limit the generalizability of the model. Research involving data from different demographic groups (e.g., age, gender, ethnicity) and geo-graphic regions could improve the model's generalizability and test its validity in a broader patient population. Utilizing larger datasets could also help examine the effects of different health conditions on the prediction of testosterone deficiency, contributing to the development of more comprehensive strategies for clinical applications. Moreover, other techniques to address class imbalance beyond SMOTE could be investigated. For instance, various data augmentation strategies, next-generation deep learning techniques, or reinforcement learning methods might offer better performance and could be tested on different healthcare data. Finally, this study also highlights the technical and ethical challenges that must be considered when applying machine learning algorithms in clinical settings. In clinical environments, it is crucial that these algorithms produce accurate and reliable results while being user-friendly and safe for healthcare professionals to use. The relevant text was added to the conclusion section.

Reviewer 2 Report
Comments and Suggestions for Authors
The introduction should be divided into 1.1 (medical topics) and 1.2 (AI in diagnostics).
What parameters were the testosterone level predictions based on?
I think it would be an interesting task to select a set of parameters that would give a more accurate prediction of testosterone levels, as well as predict the dynamics of testosterone levels depending on age.
Author Response
Comment 1: The introduction should be divided into 1.1 (medical topics) and 1.2 (AI in diagnostics).
Reply: The requested update has been made.
Comment 2: What parameters were the testosterone level predictions based on?
Reply: Total testosterone levels are the result of a complex hormonal regulation under the influence of many biological, physiological and environmental factors. These factors can affect hormonal balance in men and women and can increase or decrease total testosterone levels. Factors such as age, lifestyle factors, obesity can affect testosterone levels. There is a high prevalence of hypogonadism in specific populations such as type 2 diabetes, metabolic syndrome, obesity (Bhasin et al., (2010)). In this study, we estimated testosterone levels with age, triglycerides (mg/dl), history of hypertension, high-density lipoprotein (mg/dl), and abdominal circumference (cm) variables (Table 1).
Comment 3: I think it would be an interesting task to select a set of parameters that would give a more accurate prediction of testosterone levels, as well as predict the dynamics of testosterone levels depending on age.
Reply: Thank you for your insightful comment. We agree that the prediction of testosterone levels, as well as their dynamics over time (e.g., age-related changes), would be an interesting extension to this study. However, we would like to highlight the limitations of the current study, which might affect the ability to predict testosterone dynamics across different age groups. This study is based on a retrospective analysis using publicly available synthetic data, which has been adapted for this specific purpose. While such a dataset offers valuable insights, it lacks the longitudinal aspect of real-world clinical data, which would be necessary to model the dynamics of testosterone levels over time, particularly with respect to age. Moreover, the dataset does not include detailed longitudinal patient records that would allow us to track testosterone levels at multiple points in time for individuals. The current dataset was carefully curated through the collaboration of expert family physicians, urologists, and endocrinologists, who reviewed the data. However, without a dynamic, real-time dataset, selecting parameters that would predict testosterone fluctuations depending on age remains a challenging task. Future studies could benefit from more granular data, including multiple age groups and longitudinal records, allowing for the identification of trends and correlations between age and testosterone levels. We acknowledge that incorporating such parameters into the model could enhance its predictive power, especially when addressing the impact of age and other demographic factors on testosterone levels. However, given the current constraints, a more accurate prediction model for age-dependent testosterone dynamics would require data that includes a broader range of variables (such as time-based measures of testosterone levels, comorbidities, lifestyle factors, and medications). As such, we suggest that future research could focus on developing a dynamic model based on richer datasets to address these aspects.
Round 2
Reviewer 2 Report
Comments and Suggestions for Authors
The authors responded to my comments and added explanations to the body of the article.